# Linking Cardiovascular Disease and Metabolic Dysfunction-Associated Steatotic Liver Disease (MASLD): The Role of Cardiometabolic Drugs in MASLD Treatment

**DOI:** 10.3390/biom15030324

**Published:** 2025-02-23

**Authors:** Marios Zisis, Maria Eleni Chondrogianni, Theodoros Androutsakos, Ilias Rantos, Evangelos Oikonomou, Antonios Chatzigeorgiou, Eva Kassi

**Affiliations:** 1Medical School, National and Kapodistrian University of Athens, Mikras Asias 75, 11527 Athens, Greece; mzisis01@gmail.com (M.Z.); heliasrantos88@gmail.com (I.R.); 2Department of Biological Chemistry, Medical School, National and Kapodistrian University of Athens, 11527 Athens, Greece; marielena.hondr@gmail.com; 3Endocrine Unit, 1st Department of Propaedeutic and Internal Medicine, Laiko Hospital, National and Kapodistrian University of Athens, 11527 Athens, Greece; 4Department of Pathophysiology, Medical School, National and Kapodistrian University of Athens, 75 Mikras Asias Str., 11527 Athens, Greece; t_androutsakos@yahoo.gr; 53rd Department of Cardiology, “Sotiria” Thoracic Diseases Hospital of Athens, University of Athens Medical School, 11527 Athens, Greece; boikono@gmail.com; 6Department of Physiology, Medical School, National and Kapodistrian University of Athens, 75 Mikras Asias Str., 11527 Athens, Greece; achatzig@med.uoa.gr

**Keywords:** anti-diabetic drugs, anti-hypertensive drugs, hypolipidemic agents, MASH, MASLD, NAFLD, resmetirom

## Abstract

The link between cardiovascular disease (CVD) and metabolic dysfunction-associated steatotic liver disease (MASLD) is well-established at both the epidemiological and pathophysiological levels. Among the common pathophysiological mechanisms involved in the development and progression of both diseases, oxidative stress and inflammation, insulin resistance, lipid metabolism deterioration, hepatokines, and gut dysbiosis along with genetic factors have been recognized to play a pivotal role. Pharmacologic interventions with drugs targeting common modifiable cardiometabolic risk factors, such as T2DM, dyslipidemia, and hypertension, are a reasonable strategy to prevent CVD development and progression of MASLD. Recently, a novel drug for metabolic dysfunction-associated steatohepatitis (MASH), resmetirom, has shown positive effects regarding CVD risk, opening new opportunities for the therapeutic approach of MASLD and CVD. This review provides current knowledge on the epidemiologic association of MASLD to CVD morbidity and mortality and enlightens the possible underlying pathophysiologic mechanisms linking MASLD with CVD. The role of cardiometabolic drugs such as anti-hypertensive drugs, hypolipidemic agents, glucose-lowering medications, acetylsalicylic acid, and the thyroid hormone receptor-beta agonist in the progression of MASLD is also discussed. Metformin failed to prove beneficial effects in MASLD progression. Studies on the administration of thiazolinediones in MASLD suggest effectiveness in improving steatosis, steatohepatitis, and fibrosis, while newer categories of glucose-lowering agents such as GLP-1Ra and SGLT-2i are currently being tested for their efficacy across the whole spectrum of MASLD. Statins alone or in combination with ezetimibe have yielded promising results. The conduction of long-duration, large, high-quality, randomized-controlled trials aiming to assess by biopsy the efficacy of cardiometabolic drugs to reverse MASLD progression is of great importance.

## 1. Introduction

Non-alcoholic fatty liver disease (NAFLD) is an umbrella term used to describe disorders characterized by hepatic steatosis (histologically or radiologically identified) in the absence of excessive alcohol consumption and other secondary causes of hepatic fat accumulation [1,2]. NAFLD encompasses a wide spectrum ranging from simple hepatic steatosis to nonalcoholic steatohepatitis (NASH), which may advance to liver fibrosis and cirrhosis with liver failure and/or hepatocellular carcinoma [3,4]. It is the most common liver disease in Western countries, with approximately a global prevalence of 30% in adults, causing a constantly increasing clinical and economic burden worldwide [5]. In 2020, the entity was renamed to metabolic-associated fatty liver disease (MAFLD), underlining the metabolic substrate of the disorder, which defines its pathophysiology. MAFLD incorporates a set of “positive” criteria regardless of alcohol consumption or other concurrent liver diseases. The requirements for the diagnosis of MAFLD include evidence of hepatic steatosis (based on histology, imaging, or blood biomarkers) in addition to one of the following three criteria: overweight/obesity, presence of type 2 diabetes mellitus (T2DM), or evidence of metabolic dysregulation [6]. Therefore, MAFLD is generally considered the hepatic manifestation of the metabolic syndrome [7]. Lately, the terminology of metabolic dysfunction-associated liver disease (MASLD) was introduced to describe the previously called NAFLD entity. MASLD diagnostic criteria incorporate the presence of liver steatosis with the existence of at least one cardiometabolic risk factor, such as increased body mass index (BMI), fasting serum glucose, plasma triglycerides, blood pressure, and HDL cholesterol, in the absence of any other obvious cause of hepatic steatosis [8].

The pathophysiology of the disease has yet to be completely elucidated due to the complex mechanisms involved. The first proposed pathogenesis, termed the «two-hit hypothesis», included the consumption of a high-fat diet and the embracement of a sedentary lifestyle, leading to obesity and insulin resistance (IR), resulting in the accumulation of triglycerides (TAGs) in the liver (first hit) and prompting the expression of pro-inflammatory cytokines, leading to fat accumulation and necroinflammation (second hit) [9]. Nowadays, the “multiple-hit hypothesis” has been proposed, according to which dietary, genetic, epigenetic, and environmental factors cause obesity, IR, gut microbiome alterations, ectopic fat accumulation, adipose tissue dysfunction, dysregulation of autophagy, and mitochondrial function, leading to oxidative and endoplasmic reticulum (ER) stress and the production of reactive oxygen species (ROS) [10].

The prevalence and incidence of cardiovascular disease (CVD) and fatal cardiovascular adverse events are higher in patients with MASLD, with recent evidence suggesting that MASLD is an independent risk factor for CVD occurrence [11]. Furthermore, CVD is the most common cause of death in these patients, surpassing liver complications-related deaths [11,12].

In the majority of MASLD patients, several conventional cardiovascular risk factors are identified. Therefore, these patients are candidates for therapeutic interventions that target the reduction in the high cardiovascular burden while simultaneously having a positive effect on MASLD [13].

This review provides current knowledge on the epidemiologic association of MASLD to CVD mortality and morbidity and enlightens the possible underlying pathophysiologic mechanisms linking MASLD with CVD. It also highlights potential common therapeutic interventions with cardiometabolic drugs such as anti-hypertensive drugs, hypolipidemic agents, glucose-lowering medications, acetylsalicylic acid, and the thyroid hormone receptor-beta agonist that may improve outcomes of MASLD.

## 2. Epidemiologic Association Between MASLD and CVD Incidence and Mortality

Cardiovascular disease (CVD) encompasses a wide range of conditions that affect the heart and blood vessels [14]. The primary components of CVD are Coronary Artery Disease (CAD), Cerebrovascular Disease (CV) and Peripheral Artery Disease (PAD). CAD can manifest as acute coronary syndrome (ACS), which in turn includes myocardial infarction (MI) and unstable angina (UA), or as chronic coronary syndromes (CCS). CV consists of ischemic stroke (IS) and transient ischemic attack (TIA), as well as carotid artery stenosis. PAD affects the blood vessels located outside the heart and brain.

Increasing evidence shows a strong correlation between MASLD and CVD. MASLD seems to be independently correlated with cardiovascular adverse events even after adjustment for modifiable cardiovascular risk factors such as sex, age, body mass index (BMI), hypertension, smoking, or dyslipidemia. This knowledge has been established by several systematic reviews and meta-analyses, summarized in Table 1.

A recent meta-analysis by Mantovani et al., which included 36 observational studies, examined the risk of incidence of CVD events amongst adults with and without MASLD and demonstrated that MASLD is linked to an increased long-term risk of fatal or non-fatal CVD events (pooled random-effects HR = 1.45, 95% CI 1.31–1.61), especially in patients with more advanced liver disease and higher fibrosis stage (pooled random-effects HR 2.50, 95% CI 1.68–3.72) [15]. Wu et al. reported that MASLD was correlated with an increased risk of non-fatal CVD events (HR  1.37, 95% CI: 1.10–1.72), such as CAD, hypertension, and atherosclerosis, although without a statistically significant association with CVD mortality [16]. The latter finding may be attributed to the fluctuating frequencies of cardiovascular comorbidities among the subjects of diverse studies or the implementation of varying diagnostic criteria for MASLD and MASH. However, in a large-scale prospective cohort study with 215,245 participants, MASLD was shown to be independently associated with CVD mortality (HR 1.61, 95% CI 1.42–1.82), acute myocardial infarction (AMI) mortality (HR 1.58, 95% CI 1.19–2.11), but not with stroke mortality (HR 1.18, 95% CI 0.85–1.64) [17].

The pathologic hallmark of IR is implicated in the pathogenesis of CVD and the development and progression of MASLD. A meta-analysis of 11 studies by Zhou et al. showed that MASLD patients with T2DM had a nearly twofold increase in CVD risk compared to patients without MASLD (OR 2.20; 95% CI 1.67–2.90) [18]. Moreover, a recently published nationwide population-based study with 7,796,763 participants in Korea reported that MASLD in patients with T2DM (505,763 patients) was associated with a higher incidence rate of CVD, such as myocardial infarction, ischemic stroke, and all-cause death, even in patients with mild MASLD [19].

Regarding stroke correlation with MASLD, an increased risk of ischemic stroke (OR 1.6, 95% CI 1.2–2.1) in MASLD individuals was observed in a recent meta-analysis (Bisaccia et al.) of 33 cohort studies [20]. More interestingly, a Mendelian randomization study found a causal association between MASLD and specific ischemic stroke subtypes, i.e., large artery atherosclerosis (LAA) (OR 1.065, 95% CI 1.004–1.129; *p* = 0.037) and small vessel occlusion (SVO) (OR 1.058, 95% CI 1.003–1.116; *p* = 0.037), whereas no correlation was observed for the cardioembolic subtype (OR 1.026, 95% CI 0.983–1.071; *p* = 0.243) [21].

Growing evidence has demonstrated an independent relationship between MASLD and carotid atherosclerosis. A recent meta-analysis by Wong et al., including 67,404 MASLD individuals, demonstrated a higher prevalence of subclinical atherosclerosis, assessed by carotid intima-media thickness (CIMT) (OR 2.00, 95% CI 1.56–2.56) in patients with MASLD, a result independent of other conventional cardiometabolic risk factors [22]. In another cross-sectional study of 4112 patients aged over 40 years, MASLD was independently related to an elevated CIMT, even after adjusting for the conventional cardiometabolic risk factors (OR = 1.663, 95% CI = 1.391–1.989, *p* < 0.0001) [23].

Regarding peripheral arterial disease (PAD), a recent observational study involving 101,465 Chinese adults showed that MASLD is associated with a significantly higher risk for the prevalence (OR: 1.30 95% CI 1.19–1.42, *p* < 0.001) and incidence (adjusted HR 1.67, 95% CI 1.17–2.38, *p* = 0.005) of PAD (diagnosed by ankle-brachial index, ABI), suggesting that MASLD individuals also have an increased risk of atherosclerotic CVD in peripheral arteries [24]. Interestingly, Ciardullo et al. showed that, among 3094 MASLD individuals from the NHANES (1999–2004), followed up for a median of 13 years, PAD was associated not only with a significantly higher incidence of all-cause death (adjusted HR 1.8, 95% CI 1.4–2.4) but also with an increased risk of CVD mortality (adjusted HR 2.5, 95% CI 1.5–4.3) [25].

A significant association between MASLD and CKD has been reported by various studies [26,27,28]. A recent meta-analysis from Agustanti et al., including 355,886 patients with MASLD, followed up for 4.6–6.5 years, showed that MASLD was significantly associated with a higher prevalence (OR 1.50, 95% CI 1.02–2.23; *p* = 0.04) and incidence (adjusted HR 1.35, 95% CI 1.18–1.52; *p* < 0.001) of CKD [29]. Interestingly, the latest data also suggest that non-obese MASLD patients experience a comparable high risk of CKD development as obese MASLD individuals [30]. MASLD has also been linked to worsening CKD prognosis. Data analysis from 337,783 UK Biobank participants, followed up over a median of 12.8 years, demonstrated that MASLD was related to a two-fold increase in the risk of developing end-stage renal disease (ESRD) (HR 2.03, 95% CI 1.68–2.46; *p* < 0.001) [31]. With respect to MASLD severity, a link between hepatic fibrosis and CKD severity was observed in this study [31]. Moreover, a large Korean population-based retrospective cohort study including 816,857 individuals with an estimated glomerular filtration rate of 15–59 mL/min/1.73 m2, followed up for a median time of 7.7 years, found that a higher fatty liver index (FLI), which is a surrogate marker of MASLD, was significantly correlated with a progressively increased risk of cardiovascular and kidney major adverse events [32]. Lastly, among 28,990 Japanese individuals followed up for a mean of 6.9 years, it has also been reported that the co-existence of MASLD and CKD is an independent predictor of ischemic heart disease (HR 1.51, 95% CI 1.02–2.22) [33].

In brief, the latest epidemiologic data demonstrate an independent association of MASLD with the manifestation of CVD and/or adverse cardiovascular events. Importantly, patients with more severe liver disease and fibrosis and patients with MASLD and T2DM appear to have an increased risk for CV adverse events. While MASLD is linked to increased all-cause mortality, its association with CVD mortality remains a controversial topic.

**Table 1 biomolecules-15-00324-t001:** Meta-analyses evaluating the association of MASLD and CVD events.

Author	Study Design	Population	Results
Mantovani et al. (2021) [15]	Systematic review and meta-analysis	36 observational studies 5,802,226 adults 335,132 individuals with baseline MASLD (diagnosed with liver biopsy, imaging techniques, ICD 9/10 codes) Median follow-up period: 6.5 years	Increased risk of fatal or non-fatal CVD events in patients with MASLD vs. those without (pooled random-effects HR = 1.45, 95% CI 1.31–1.61). Even higher risk with more severe MASLD (pooled random-effects HR = 2.50, 95% CI 1.68–3.72). All risks were independent of other common cardiometabolic risk factors.
Wu et al. (2016) [16]	Systematic review and meta-analysis	34 studies (21 cross-sectional studies, and 13 cohort studies) 164,494 participants 153,209 patients with MASLD (diagnosed by U/S, CT or liver biopsy)	Increased risk of prevalence (OR = 1.81, 95% CI: 1.23–2.66) and incidence (HR = 1.37, 95% CI: 1.10–1.72) of CVD in patients with MASLD vs. those without MASLD. Increased risk of prevalence (OR = 1.87, 95% CI: 1.47–2.37) and incidence (HR = 2.31, 95% CI: 1.46–3.65) coronary artery disease (CAD), prevalence (OR = 1.24, 95% CI: 1.14–1.36) and incidence (HR = 1.16, 95% CI: 1.06–1.27) of hypertension and prevalence (OR = 1.32, 95% CI: 1.07–1.62). atherosclerosis among patients with MASLD than those without MASLD. No statistically significant difference in CVD mortality between patients with MASLD and non-MASLD participants (HR = 1.10, 95% CI: 0.86–1.41).
Bisaccia et al. (2023) [20]	Systematic review and meta-analysis	33 cohort studies 10,592,851 individuals 219,211 patients with MASLD (diagnosed by U/S, CT or liver biopsy) Mean follow-up time: 10 ± 6 years	Increased risk of MI (OR = 1.6, 95% CI: 1.5–1.7, *p* < 0.001), stroke (OR = 1.6, 95% CI, 1.2–2.1, *p* = 0.005), atrial fibrillation (OR = 1.7, 95% CI, 1.2–2.3, *p* = 0.001), and major adverse cardiovascular and cerebrovascular events (OR: 2.3, 95% CI, 1.3–4.2, *p* = 0.01) among patients with MASLD than those without MASLD. No statistically significant association between MASLD and CVD mortality (OR= 1.13, 95% CI, 0.57–2.23; *p* = 0.656) or all-cause mortality (OR= 1.14, 95% CI, 0.78–1.67, *p* = 0.459) between MASLD and non-MASLD patients.
Targher et al. (2016) [34]	Meta-analysis	16 observational studies 34,043 adults 36.3% of individuals with baseline MASLD Median follow-up period: 6.9 years	Increased risk of fatal and/or non-fatal CVD events in patients with MASLD vs. those without MASLD (random effect OR = 1.64, 95% CI 1.26–2.13). Even higher risk in patients with more severe MASLD (random effect OR = 2.58; 95% CI 1.78–3.75).
Abosheaishaa et al. (2024) [35]	Systematic review and meta-analysis	32 studies 5,610,990 individuals 567,729 patients with MASLD	Increased risk of angina (RR = 1.45, 95% CI: 1.17–1.79), CAD (RR = 1.21, 95% CI: 1.07–1.38), Coronary artery calcium (CAC) > 0 (RR = 1.39, 95% CI: 1.15–1.69), and calcified coronary plaques (RR = 1.55, 95% CI: 1.05–2.27). No statistically significant association between MASLD and CAC >100 (RR = 1.16, 95% CI: 0.97–1.38) and MI (RR = 1.70, 95% CI: 0.16–18.32).
Liu et al. (2019) [36]	Meta-analysis	14 studies 498,501 individuals More of 95,111 patients with MASLD	Increased risk of all-cause mortality in patients with MASLD vs. those without (HR = 1.34, 95% CI 1.17–1.54). No statistically significant association of MASLD with CVD mortality (HR = 1.13, 95% CI 0.92–1.38).

## 3. Pathophysiological Linkage of MASLD and CVD

The underlying pathophysiologic linkage of MASLD and CVDs involves many complex, frequently overlapping pathways [37].

### 3.1. Dyslipidemia

Dyslipidemia includes a broad spectrum of lipid perturbations manifesting as increased plasma TG and low-density lipoprotein (LDL) cholesterol and decreased levels of high-density lipoprotein (HDL) cholesterol.

Lipid metabolism is regulated by the liver via the combined action of de novo lipogenesis (DNL) and breakdown of lipids, as well as the uptake and secretion of serum lipoproteins [38]. As hepatic fat accumulation is the hallmark of MASLD, it is clear why impaired lipid metabolism is identified as one of the major risk factors of the disease. Impaired liver uptake of serum lipids, increased hepatic DNL, and altered export of very low-density lipoproteins (VLDL) are observed in MASLD [39], leading to the accumulation of free fatty acids (FFA) in liver parenchyma and thus the development of steatohepatitis (NASH).

Overproduction of VLDL and LDL, in combination with low serum HDL levels contribute to the phenotype of atherogenic dyslipidemia that consists of a predominance of particularly atherogenic, small dense LDL particles, which in turn initiates the cascade of atherosclerosis by penetration and accumulation of apolipoprotein-B-containing lipoproteins within the subendothelial layer of vessels [40,41]. LDL cholesterol in the vascular wall is further oxidized and stimulates the innate immune response via toll-like receptors (TLRs) [42]. Apo-B-containing particles play a crucial role in atherosclerosis development. Moreover, triglyceride-rich lipoproteins such as VLDL and intermediate-density lipoprotein (IDL) promote CVD development. In these particles, apolipoprotein C3 is contained, which likely plays a role in activating the inflammasome [43], which, in turn, activates the interleukin (IL)-1β family via caspase-1, stimulating pro-inflammatory mediators IL-1, IL-6, and CRP in the vascular wall and promoting vascular inflammation and atherosclerotic CVD [44]. Likewise, increased hepatic DNL in MASLD is also associated with an elevated production of saturated fatty acids, such as hepatic palmitic acid (C 16:0), which, in turn, can induce vascular inflammation by inflammasome activation [45].

### 3.2. Inflammation–Oxidative Stress

Liver inflammation in MASLD can be considered a multifactorial process, in which inflammatory stimuli may derive from either hepatic tissue or extrahepatic regions, such as the adipose tissue and the gut [46]. Circulating levels of pro-inflammatory cytokines [IL-6, high sensitivity CRP, IL-1β, Tumor Necrosis Factor-a (TNF-α)] are often increased in patients with MASLD, especially in MASH, indicating that low-grade systemic inflammation is a major component in disease progression [47]. This systemic inflammation can, in turn, lead to endothelial dysfunction and atherosclerosis progression and is associated with CVD [40].

Hepatocyte-derived extracellular vesicles (EVs) can trigger endothelial inflammation via microRNA-1 (miR-1), a critical component within EVs that suppresses Krüppel-like factor 4 (KLF4) expression and activates the nuclear factor kappa-B (NF-κB) pathway [48]. The hepatic miRNAs can subsequently enter the bloodstream and trigger CVD through modifications in lipid metabolism and/or the induction of systemic inflammation [49].

In patients with MASLD, changes in methionine metabolism and subsequent alternation of homocysteine catabolism are implicated in the increased levels of serum homocysteine that are regularly seen [50]. Elevated serum levels of homocysteine can deplete glutathione stores, leading to oxidative stress, and thus increased vascular resistance and impaired nitric oxide formation [51,52]. Superoxide-induced oxidative stress may also cause endothelial dysfunction, a crucial early step in atherosclerosis formation [53].

### 3.3. Insulin Resistance (IR)

As already mentioned, IR has now been identified as one of the major drivers of hepatic steatosis in patients with MASLD, as well as a major risk factor for CVD. Systemic inflammation, visceral obesity, and augmented accumulation of dysfunctional ectopic adipose tissue lead to IR [54].

Excessive intrahepatic accumulation of lipid metabolites, such as diacylglycerol, has been implicated as a mediator of IR [55]. Visceral adipose tissue also contributes to MASH development by secreting several adipokines and pro-inflammatory molecules such as adiponectin, leptin, resistin, TNF-α, IL-1β, and IL-6 that, in turn, downregulate glucose transporter GLUT-4 expression [56].

It should be highlighted that IR results in compensatory persistent hyperinsulinemia, which is of crucial importance for the development of unfavorable metabolic events [57]. Both increased circulating glucose levels and hyperinsulinemia stimulate DNL in MASLD. One of the underlying mechanisms of DNL is the activation of the sterol regulatory element-binding protein 1c (SREBP-1c) and carbohydrate-responsive element-binding protein (ChREBP) transcription factors, which induce the expression of enzymes involved in lipogenesis [58]. It has been observed that DNL contributes to the synthesis of 26% of intrahepatic triglycerides in MASLD patients, compared to 5% in healthy individuals [59].

Impaired insulin signaling has deleterious effects on the endothelium and vascular smooth muscle cells, enhancing endothelial dysfunction and the progression of atherosclerosis. Elevated insulin levels overactivate the phosphoinositide 3-kinase/protein kinase B (PI3K-PKB) pathway, enhancing the development or progression of atherosclerosis [60]. Endothelial insulin resistance leads to increased expression of intracellular adhesion molecule 1 (ICAM-1) and vascular cell adhesion molecule 1 (VCAM-1) due to the downregulation of the insulin receptor–Akt1 pathway [61]. The activation of endothelial nitric oxide (NO) synthase in endothelial cells is also impaired, resulting in reduced production of NO. Vascular smooth muscle cells undergo phenotypic alternation, which may participate in atherosclerosis progression [60]. Oxidative stress and concurrent activation of inflammatory pathways seem to be triggered by persistent hyperglycemia and high postprandial glucose levels, leading to the production of advanced glycation end products (AGEs) that also promote vascular inflammation. Moreover, IR is associated with dysregulated activation of the renin–angiotensin–aldosterone system (RAAS) and elevated levels of plasminogen activator inhibitor-1 (PAI-1) that lead to dysfunction of the fibrinolytic system, as well as the development and progression of autonomic neuropathy, which could promote myocardial systolic and diastolic dysfunction or life-threatening arrhythmias [61,62,63].

### 3.4. Hepatokines

Hepatokines are proteins produced by liver cells that can influence metabolic processes through autocrine, paracrine, and endocrine ways. Key examples of this group of proteins include fetuin-A, fibroblast growth factor 21 (FGF21) and angiopoietin-like proteins (ANGPTLs). Hepatic steatosis, the hallmark of MASLD, may induce changes in the secretion of hepatokines [64]. Some of these proteins contribute to the further development of MASLD by inducing oxidative stress, promoting systemic inflammation, and modulating glucose and lipid metabolism, while others have a negative regulatory effect. Fetuin-A, a glycoprotein mainly produced and excreted by hepatocytes, acts as a tyrosine kinase inhibitor of insulin receptors in the liver. From a pathogenetic aspect, high Fetuin-A levels promote MASLD development by inducing insulin resistance and a low-grade, subclinical inflammatory response [65,66,67]. Furthermore, it has been reported in observational studies that Fetuin-A circulating levels are increased in MASLD patients, and elevated serum Fetuin-A is an independent predictor of MASLD occurrence [68]. Most interestingly, Fetuin-A levels are also positively related to T2DM, myocardial infarction, and ischemic stroke, evidence that might render it a possible causative link between NASH and CVD [69,70,71]. However, higher levels of Fetuin-A may also have a protective role in nondiabetic older patients with CVD [72]. Fibroblast growth factor 21 (FGF-21), a member of the fibroblast growth factor family, is a hepatokine that regulates glycemic control and energy expenditure [73]. FGF21 has been reported to promote glucose uptake in adipose tissue, increase insulin secretion, improve insulin sensitivity, and reverse hepatic steatosis [74]. Increased serum levels of FGF-21 have also been shown to be an independent predictor of MASLD [75]. As regards its emerging role in CVD, the application of FGF-21 treatment in prediabetic rats ameliorates insulin resistance and provides cardio-protection in multiple ways, including signaling pathways related to mitochondrial fatty acid oxidation, as well as the upregulation of anti-apoptotic genes [76]. ANGPTLs are a family of secretory glycoproteins with high homology to the angiopoietins, which are vastly expressed in the liver. Up to now, eight ANGPTLs have been discovered, from ANGPTL1 to ANGPTL8. ANGPTLs have been reported to function in several biological processes such as glucose metabolism, lipid trafficking, and angiogenesis [77]. Studies suggest that ANGPTL3, ANGPTL4, and ANGPTL8 have major roles in the pathogenesis of MASLD via the regulation of vitamin D receptor (VDR), IR, and lipid metabolism. JAK2/STAT3 and PI3K/Akt signaling seem to be important molecular pathways involved in ANGPTL-mediated pathogenetic processes leading to MASLD [78]. A recent meta-analysis of 13 studies found that MASLD patients had higher circulating levels of ANGPTL8 than the healthy controls [79]. On the contrary, ANGPTL3 levels are increased only in patients with MASH [80]. In addition to that, emerging evidence demonstrates that ANGPTL proteins are involved in the pathology of the atherosclerotic process, causing endothelial dysfunction, dyslipidemia, inflammation, and platelet activation [81]. Serum ANGPTL2 has been shown to be an independent risk biomarker for CVD in T2D patients, with elevated ANGPTL2 levels associated with a 1.18-fold increased risk of CVD [82]. Moreover, circulating ANGPTL8 levels have also been identified as an independent risk factor for CAD in nondiabetic Chinese individuals [83].

### 3.5. Genetics

Apart from the environmental factors, the causal relationship between MASLD and CVD depends on a genetic background with several genetic polymorphisms co-existing in both NAFLD and CVD patients [84]. Of note, patatin-like phospholipase domain-containing protein 3 (PNPLA3) and transmembrane 6 superfamily member 2 (TM6SF2) are thought to have a protective effect regarding CVD risk, although positively related to MASLD. In particular, PNPLA3 is a multifunctional enzyme that participates in lipid droplet remodeling and triglyceride metabolism [85], while TM6SF2 regulates hepatic VLDL excretion [86]. Robust data show that the *PNPLA3* rs738409 polymorphism predisposes to MASLD and is linked to an increased risk of developing MASH, advanced fibrosis, and cirrhosis [87]. The *TM6SF2* rs58542926 polymorphism has also been associated with MASLD predisposition [88]. On the other hand, an exome-wide association study of plasma lipids in more than 300,000 participants determined that these alleles seem to confer modest protection from CAD by lowering serum triglycerides and LDL cholesterol levels [89]. However, to what extent these genetic modifications can also affect the development of other cardiovascular diseases is still under debate.

### 3.6. Gut Dysbiosis

Alterations in gut microbiota, referred as intestinal dysbiosis, have been linked to the development of MASLD and several cardiometabolic diseases such as atherosclerosis and hypertension [90,91]. The gastrointestinal tract can be considered as a possible site of origin of systemic inflammation that may play a crucial role in the establishment and maintenance of metabolic diseases such as MASLD and CVD. Gut dysbiosis is associated with an impairment in intestinal barrier function leading to increased mucosal barrier permeability to intestinal bacteria or microbial-derived products that, in turn, induce a systemic inflammatory response [57].

Secondary bile acids, trimethylamine (TMA), and short-chain fatty acids are a few of the described microbial-derived metabolites [92], which, via modulation of energy balance and insulin sensitivity, but also through triggering adipose inflammation, may have an impact on MASLD and CVD development. TMA N-oxide (TMAO), a TMA metabolite, also triggers platelet activation and its plasma levels are linked to thrombotic event risk [93]. The introduction of gut microbiota with pro-inflammatory properties into Ldlr−/− female mice has also been shown to accelerate atherosclerosis and increase systemic inflammation [94]. In this context, the use of statins has been associated with a lower prevalence of gut microbiota dysbiosis through the well-established anti-inflammatory effects of these drugs [95].

In line with the aforementioned data, a variety of upcoming therapeutic options focus on targeting the gut–liver axis. Obeticholic acid is a synthetic variant of the natural bile acid chenodeoxycholic acid which acts as an agonist on the farnesoid X receptor in intestinal epithelial cells and hepatocytes, regulating bile acid, glucose, and lipid metabolism [96]. In the FLINT clinical trial, a randomized, placebo-controlled study of 283 patients with MASH, an improvement in the hepatic histological parameters was shown in patients receiving obeticholic acid [97]. However, the proportion of patients with resolution of MASH did not differ between obeticholic acid and placebo group. Obeticholic acid was also associated with increased total serum and LDL cholesterol and reduced HDL cholesterol, leading to an increased CVD risk profile.

### 3.7. Other Potential Mechanisms

Growing evidence suggests that the increase in epicardial adipose tissue can be considered a potential cardiometabolic risk factor [98]. MASLD is associated with increased epicardial adipose tissue, and, simultaneously, a higher epicardial fat thickness is associated with more severe liver fibrosis, steatosis, and CVD prevalence in MASLD patients [99]. Under physiological conditions, epicardial fat supplies energy and heat to the myocardium and exerts an anti-inflammatory, antioxidative, and anti-atherogenic cardioprotective role. A co-occurrence of increased epicardial fat thickness and other metabolic and hemodynamic abnormalities converts epicardial fat into a highly lipotoxic, prothrombotic, profibrotic, and pro-inflammatory organ [100], leading to the release of pro-inflammatory cytokines (e.g., leptin, TNF-a, IL-1, IL-6, and resistin), that in turn promote fibrosis and infiltration of the coronary intima wall layer with macrophages, and thus, accelerate atherosclerosis [101]. Regarding liver tissue, the released pro-inflammatory cytokines may contribute to the activation of hepatic stellate cells and induce hepatic fibrosis.

Several pro- and anticoagulant factors are mainly produced in the liver. An increase in coagulation factor VIII, IX, XI, and XII activities has been described in MASLD, proposing that hepatic fat accumulation can increase thrombotic risk [102]. Likewise, plasminogen inhibitor activator 1 (PAI-1) serum levels are also positively related to liver fat content [103]. Figure 1 depicts the proposed pathophysiologic mechanisms involved in the causal linkage between MASLD and CVD.

## 4. Cardiometabolic Drugs and MASLD

The strong association between CVD and MASLD emphasizes the need for early identification and adequate treatment of cardiometabolic risk factors. Current guidelines suggest that all patients with MASLD should be screened regularly to detect the presence of additional cardiometabolic risk factors such as dyslipidemia, hypertension, obesity, or T2DM [2]. Until recently, there were no specific pharmacological therapies for MASLD; therefore, optimal management of cardiometabolic risk factors implicated in the pathophysiology of the disease is crucial to reduce the cardiovascular risk of these patients. The recent approval of resmetirom, a thyroid hormone receptor b (THRb), opens new opportunities in the therapeutic management of MASLD [104].

### 4.1. Anti-Hypertensive Drugs

Among MASLD patients, the prevelence of arterial hypertension ranges from 40 to 70% and emerging data from prospective studies highlight the strong linkage between MASLD and an increased risk of pre-hypertension (i.e., systolic blood pressure: 120–139 mmHg, diastolic blood pressure: 80–89 mmHg) and hypertension [105,106].

In spite of the significant correlation between MASLD and arterial hypertension, current evidence on the relative advantage of certain anti-hypertensive agents in metabolic dysfunction-associated steatohepatitis (MASH) is insufficient. Therefore, anti-hypertensive management in patients with MASLD is conducted following current guidelines for arterial hypertension irrespective of this diagnosis.

#### 4.1.1. Renin–Angiotensin–Aldosterone System (RAAS) Inhibitors

Renin–angiotensin–aldosterone system (RAAS) is a major contributor to the homeostasis of extracellular volume and blood pressure. Simultaneously, RAAS overactivity leads to a cascade of deleterious changes in the cardiovascular system promoting endothelial dysfunction, vascular smooth muscle cell and monocyte proliferation, pro-thrombotic effects, and IR [107].

Currently, Angiotensin Converting Enzyme (ACE) inhibitors and Angiotensin Receptor Blockers (ARBs) are used in several CVD scenarios, such as in patients with hypertension, for secondary prevention after myocardial infarction, as well as for the management of heart failure with reduced ejection fraction (HFrEF), and their cardiovascular benefit has already been well-established [108,109].

In preclinical mice studies, RAAS inhibitors have been shown to have a positive effect on liver fibrosis, fat deposition, and necroinflammation through variable mechanisms [110,111,112]. Furthermore, telmisartan has been shown to work as a partial peroxisome proliferator-activated receptor gamma (PPAR-γ) agonist, influencing the expression of PPAR-γ target genes involved in carbohydrate and lipid metabolism and reducing glucose, insulin, and triglyceride levels, thus reducing IR [113].

In a pilot study of patients with MASLD and chronic hepatitis C receiving telmisartan or olmesartan, an improvement in IR and transaminase levels was observed [114]. Moreover, in a retrospective cohort study including more than 12,000 patients with MASLD, treatment with ACEIs was correlated with a reduced risk of cirrhosis and liver cancer, even though this effect was not consistent with ARB treatment [115]. The effects of telmisartan or losartan for MASLD treatment were studied in the FANTASY trial, which included 19 hypertensive patients with MASLD. Although serum FFA levels were significantly reduced in the telmisartan compared to the losartan group, there was no significant improvement in liver function in either group [116]. Furthermore, even though telmisartan has been shown to improve NAS and fibrosis scores in patients with MAS [117], a recent meta-analysis from Li et al. failed to prove a statistically significant decrease in serum ALT levels (although demonstrating a decreasing trend) with neither improvement in fibrosis nor NAFLD scores with ARB therapy [118]. Therefore, the current evidence from clinical trials is insufficient to support the efficacy of RAAS inhibitors in MASLD patients.

#### 4.1.2. Mineralocorticoid Receptor Antagonists (MRAs)

The harmful effects of aldosterone on the cardiovascular system have been previously well described, including sodium and fluid retention, myocardial fibrosis, and endothelial dysfunction [119]. MRAs are currently used as second-line anti-hypertensive agents in patients with resistant hypertension [120]. Furthermore, MRAs are part of the quadruple therapy for HFrEF that improves survival and reduces hospitalization for HF [121]. Three landmark randomized controlled trials (RCTs) have shown the benefits of MRAs in decreasing morbidity and mortality among patients with reduced left ventricle (LV) ejection fraction (RALES trial, EPHESUS trial, EMPHASIS trial) [122,123,124]. In heart failure with preserved ejection fraction (HfpEF), MRAs have been shown to improve echocardiographic parameters of diastolic function [125], with no reduction in the risk of major cardiovascular adverse effects [126].

In experimental mouse models, it has been shown that the expression of MR is associated with hepatic IR, liver inflammation, and fibrosis development, while MR blockade with eplerenone induces anti-steatotic and anti-fibrotic effects [127,128]. Furthermore, reduced activation of hepatic MR seems to lead to beneficial downstream inhibition of lipogenesis [129]. Unfortunately, data from clinical trials demonstrating possible beneficial effects of MR inhibition in patients with MASLD are largely lacking. A randomized, double-blind, placebo-controlled trial (MIRAD trial) studying eplerenone’s effect on liver fat and metabolism in patients with T2DM did not show promising beneficial effects [130]. A combination of spironolactone with vitamin E, on the other hand, appeared to have a positive effect on serum insulin levels and HOMA-IR in MASLD patients [131]. Moreover, Polyzos et al., in their 52-week randomized controlled trial including 23 women with MASLD, compared the effects of combined low-dose spironolactone plus vitamin E vs. vitamin E monotherapy on MASLD. They showed that the NAFLD liver fat score, insulin levels, and HOMA-IR declined significantly only in the combination treatment group [132]. Overall, the role of MRAs in MASLD treatment is still under debate.

#### 4.1.3. Calcium Channel Blockers

Calcium channel blockers are currently one of the first options in the management of arterial hypertension [133].

In a recent preclinical study by Li et al., it was shown that, in mice with MASLD and hypertension, amlodipine, via modulation of gut microbiota, ameliorates liver injury and steatosis while improving lipid metabolism through a reduction in the expression of lipogenic genes [134]. Likewise, the use of amlodipine in spontaneously hypertensive rats with steatohepatitis improved IR, and cytokine profile of IL-6 and IL-10, suggesting that amlodipine could be beneficial for patients with metabolic syndrome and MASH [135]. However, there is a lack of clinical data determining if calcium channel blockers are effective for the prevention of CVD in patients with MASLD.

#### 4.1.4. Beta Blockers

The maladaptive activation of SNS is implicated in the pathophysiology of several cardiovascular diseases (e.g., heart failure [136]). Beta-blockers exert their cardioprotective effects through several mechanisms (e.g., anti-inotropic, anti-chronotropic action), rendering them crucial therapeutic choices in patients with CVD. Up to date, they are used for the relief of angina pectoris symptoms, as anti-hypertensive drugs in selected patients, after acute coronary syndromes, and in combination with RAS inhibitors, MRAs, and SGLT2i in HFrEF [121,137,138,139].

In a preclinical MASH rat model, propranolol was shown to worsen liver injury via activation of apoptotic pathways, suggesting that beta-blockers should be avoided or used with extreme caution in patients with MASH [140]. Studies evaluating the effect of this drug category in humans have yet to be conducted.

### 4.2. Anti-Hyperglycemic Agents

Given the close relationship between IR, MASLD, and CVD, it is not a surprise that fifty percent of all NAFLD patients have T2DM, while the prevalence of MASLD is higher in individuals with prediabetes [141]. Furthermore, T2DM is closely associated with the severity of MASLD and promotes the development of MASH, advanced fibrosis, and HCC [142,143]. Conversely, patients with U/S-defined MASLD have a 2–5-fold increased risk of subsequent incident T2DM [144]. Therefore, diabetic patients should be assessed for possible MASLD, and vice versa, screening and surveillance for T2DM should be performed in patients with MASLD [2].

#### 4.2.1. Glucagon-like Peptide-1 Receptor Agonists (GLP-1RAs)

Glucagon-like peptide 1 (GLP-1) is a polypeptide secreted by intestinal L-cells after meal digestion. GLP1-RAs act by enhancing glucose-mediated insulin release and suppressing glucose-mediated glucagon secretion in pancreatic tissue [145]. GLP-1RAs are used in clinical practice as antidiabetic drugs. Furthermore, the ability of these drugs to reduce appetite, acting as central anorexigens and delaying gastric emptying, makes them suitable for weight reduction in obese patients in whom non-pharmacologic interventions have failed to reduce body weight [146].

GLP-1RAs have demonstrated unique properties beyond glucose metabolism regulation, exerting cardioprotective and vasodilatory effects [147]. The cardiovascular benefits of GLP-1RAs have been shown in several RCTs among patients with T2DM and high cardiovascular risk [148,149,150,151]. Recently, semaglutide, a drug of this category, showed a reduction in major adverse cardiovascular events in overweight or obese patients without T2DM [152].

In hepatic cells, through the activation of AMP kinase, GLP1-RAs reduce liver lipogenesis [153]. In vitro, they have been shown to decrease triglyceride stores and steatosis in human hepatocytes via modulation of insulin signaling pathways, as well as hepatic inflammation, via reducing fat-derived oxidants [154,155]. Data from clinical studies demonstrate a positive effect of GLP-1RAs in patients with MASLD [156]. Most trials have shown an improvement in serum aminotransferases and liver steatosis on imaging [157,158,159,160]. Nevertheless, few RCTs have incorporated paired liver histological outcomes. The latter studies have demonstrated NASH resolution without improvement in liver fibrosis stage [161,162,163].

A promising meta-analysis of eleven placebo-controlled or active-controlled phase-2 RCTs demonstrated that treatment with GLP1-RAs (exenatide, semaglutide, liraglutide, dulaglutide) in patients with MASLD was correlated with a reduction in serum liver enzyme levels and an absolute percentage of liver fat content on magnetic resonance-based techniques and a histological resolution of MASH without worsening of liver fibrosis. However, improvement in the liver fibrosis stage was not observed in either of these trials [164].

Of note, a paired-biopsy phase 3 clinical trial (NCT04822181) with 1200 participants with NASH and significant fibrosis (stages F2–F3) receiving semaglutide 2.4 mg/week or placebo for 72 weeks is currently ongoing.

GLP-1RAs, therefore, could be considered a drug of choice for patients with MASLD, especially in obese patients with T2DM and MASH. It is worth noting that, until now, data regarding the use of GLP-1RAs in MASH patients with no T2DM are scarce.

#### 4.2.2. Dual Glucagon-like Peptide-1/Glucose-Dependent Insulinotropic Peptide (glp-1/gip) Receptor Agonist

Tirzepatide is a novel dual GLP1-GIP receptor agonist that has been recently shown to induce substantial and sustained body weight loss in overweight patients with or without diabetes mellitus (SURMOUNT TRIALS) [165,166]. In May 2022, tirzepatide received FDA approval to improve glycemic control in adults with T2DM [167].

The cardiovascular outcomes of tirzepatide in patients with T2DM and increased CV risk are currently being studied in the SURPASS-CVOT trial [168], while the drug’s potential cardiovascular benefits in overweight or obese patients with established CV disease but without diabetes mellitus are being evaluated in the SURMOUNT-MMO trial.

In SURPASS-3 MRI, a substudy of the SURPASS-3 trial, tirzepatide promoted a significant reduction in liver fat content (LFC) and visceral adipose tissue compared with insulin degludec [169]. In a proof-of-concept trial studying tirzepatide’s effect on MASH activity and fibrosis biomarkers in patients with T2DM, higher tirzepatide doses significantly increased adiponectin levels and reduced MASH-related biomarkers [170]. Given these biomarker effects, the ongoing SYNERGY-NASH trial aims to demonstrate the clinical effect of tirzepatide by measuring MASH resolution and lack of fibrosis progression. In summary, the effects of tirzepatide with its dual mechanism of action and the upcoming triple agonists (GLP-1/GIP/glucagon receptor triagonist) with greater outcomes on weight loss have the potential to outperform the known effects of single GLP-1Ras and become an alluring treatment option for MASLD or MASH, especially for patients with coexisting diabetes mellitus or obesity.

#### 4.2.3. Sodium Glucose Transporter-2 Inhibitors (SGLT-2i)

Sodium glucose transporter-2 inhibitors (SGLT-2i) were initially designed as antidiabetic drugs for patients with T2D, acting on the sodium/glucose cotransporter 2 protein expressed in the renal proximal convoluted tubules of the nephron, leading to a reduction in the reabsorption of filtered glucose and thus, improving glycemic control [171,172]. Furthermore, through the stimulation of the SIRT1/AMPK signaling pathway and the downregulation of the Akt/mTOR signaling pathway, SGLT-2i can trigger a variety of glucose-independent beneficial effects: amelioration of oxidative stress, reduction in inflammation, normalization of mitochondrial structure and function, minimization of coronary microvascular injury, improvement of contractile performance, and suppression of the development of cardiomyopathy. In the nephrons, a reduction in glomerular and tubular inflammation is observed, diminishing the development of nephropathy [173]. Clinically, SGLT2i cardiovascular benefit has been proven in patients with established atherosclerotic cardiovascular disease (ASCVD) and T2DM, rendering them a first-choice therapeutic approach for this subgroup of patients [174]. Recently, SGLT-2i inhibitors have become part of the standard of care in patients with HFrEF, with clear beneficial effects in reducing the risk for CV death or hospitalization for heart failure, even in patients without T2DM [121], while the EMPEROR-PRESERVED trial has demonstrated their efficacy in CV risk reduction in patients with HFpEF [175].

In experimental animal models, SGLT-2i have been reported to restrict the development of MASLD and ameliorate histological hepatic steatosis or steatohepatitis in both weight-dependent and independent ways. They also reduce serum liver enzymes, hepatic collagen deposition, and inflammatory cytokine expression [176,177,178].

Several clinical studies highlight the beneficial effects of SGLT-2i in patients with T2DM and MASLD. In RCTs using as a primary outcome MRI-measured changes in liver triglyceride content, SGLT2 inhibitors have been shown to reduce liver steatosis [179,180,181].

In the EMPA-REG OUTCOME trial, which included pooled data from four placebo-controlled trials (n = 2477) and a trial of empagliflozin versus glimepiride, empagliflozin was shown to significantly reduce aminotransferase levels in patients with T2DM [182]. A meta-analysis of ten randomized controlled trials of a total of 573 participants with T2DM and MASLD supports that SGLT2 inhibitors are superior over other antihyperglycemic drugs used in these RCTs in improving serum aminotransferases levels, hepatic fibrosis and lowering liver fat and body weight [183]. However, data in non-diabetic patients with MASLD are lacking. There is only a small single-center study comparing 12 patients under dapagliflozin and 10 patients under teneligliptin, a DPP4 inhibitor, that showed that after an intervention period of 12 weeks, serum transaminases were reduced in both groups, while in the dapagliflozin group, total body water and body fat also decreased, leading to decreased total body weight [184].

Proposed pathophysiologic mechanisms that may lead to MASLD improvement under SGLT-2i treatment include reduced glucose and insulin levels, which in turn, lead to reduced de novo endohepatic lipid synthesis and increased glucagon secretion from alpha pancreatic cells that also express SGLT-2. Moreover, hepatic β-oxidation is stimulated by the elevated serum glucagon levels, and an anti-oxidant environment is established through a decrease in high glucose-induced oxidative stress, a reduction in free radical generation, and an upregulation of anti-oxidant systems [185].

#### 4.2.4. Metformin

Metformin, an AMPK agonist, exerts its actions by improving insulin sensitivity in peripheral tissues and regulating glucose utilization by the hepatic cells [186]. Due to its high efficacy in reducing blood glucose levels and improving insulin sensitivity, as well as its well-established safety, metformin is a first-line medical treatment in T2DM patients [186]. Apart from its glucose-lowering effect, metformin exerts pleiotropic actions in several cardiovascular parameters, including dyslipidemia, systemic inflammation, endothelial dysfunction, oxidative stress, and atherosclerosis [187].

A recent meta-analysis of 16 studies, including 701,843 participants of T2DM, treated with metformin, and 1,160,254 controls, demonstrated a reduced mortality risk [OR = 0.44 (0.34–0.57)] and adverse cardiovascular events [OR = 0.73 (0.59–0.90)] with metformin [188]. Furthermore, an ongoing trial (the “MetCool ACS” trial) is going to evaluate the effectiveness of metformin in reducing the risk of an unscheduled PCI or CABG after successful revascularization due to ACS within a 30-month follow-up period in patients without diabetes mellitus. Additionally, a meta-analysis of nine RCTs (754 patients) suggested a favorable effect of metformin concerning left ventricular mass index (LVMI) and LVEF in patients with or without preexisting CVD [189].

As expected, the efficacy of metformin in MASLD has been thoroughly investigated, both in patients with T2DM and non-diabetic individuals. Although some trials have shown a potential beneficial effect in reducing cirrhosis and HCC risk [190,191], two recent major meta-analyses failed to demonstrate the beneficial effect of metformin in improving hepatic biochemical or histological parameters of patients with MASLD [192,193]. Therefore, metformin is not recommended as a specific MASLD treatment in current guidelines from international societies [2].

#### 4.2.5. Thiazolidinediones (TZDs)

TZDs are potent PPAR-γ agonists that improve IR, a hallmark of T2DM, MASLD, and CVD [194]. They have been used as antidiabetic agents, but their adverse effects have rendered them a backup choice in patients with T2DM [194].

Apart from its glucose-lowering effects, in large clinical trials, pioglitazone seems to retard the atherosclerotic process and reduce the risk of cardiovascular events [195,196,197]. A meta-analysis of 19 trials with a total of 16,390 patients with T2DΜ receiving either pioglitazone or placebo showed that the use of pioglitazone was associated with a reduced primary composite endpoint (death, myocardial infarction, or stroke) (HR, 0.82; 95% CI, 0.72–0.94). However, serious heart failure events were more prominent in the pioglitazone group (HR, 1.41; 95% CI, 1.14–1.76), without an increase in mortality [198]. On the contrary, a meta-analysis of several small studies suggested an association of rosiglitazone with MI and an overall trend toward death from CV causes [199]. Therefore, rosiglitazone has been subsequently removed from the market.

Animal studies have demonstrated improvement in various aspects of MASLD, such as improvement of hepatic steatosis and fibrosis and suppression of inflammation [200,201].

Additionally, several clinical trials demonstrate that treatment with TZDs, and especially pioglitazone, can lead to an improvement in hepatic biochemical and histologic parameters in patients with NASH [202,203,204,205,206]. In a randomized control clinical trial of 247 MASH patients without T2DM receiving either pioglitazone (80 patients), or vitamin E (84 patients), or placebo (83 patients), the use of pioglitazone was associated with a statistically significant reduction in plasma alanine and aspartate aminotransferase levels, hepatic steatosis, and lobular inflammation when compared to the placebo group, but without improvement in fibrosis scores [207]. A meta-analysis of four randomized, placebo-controlled clinical trials using pioglitazone or rosiglitazone versus placebo in the treatment of 344 patients with MASH showed that TZDs were more likely to improve ballooning degeneration (OR 2.1, 95% CI 1.3–3.4), lobular inflammation (OR 2.6, 95% CI 1.7–4.0), steatosis (OR 3.4, 95% CI 2.2–5.3) and necroinflammation (OR 6.52, 95% CI, 3.03–14.06). Moreover, an improvement in fibrosis was observed in patients treated with pioglitazone compared with placebo (OR 1.7, 95% CI 1.0–2.8) [208]. In another meta-analysis of 8 RCTs with 516 patients with biopsy-proven MASH, pioglitazone therapy was associated with an improvement in advanced fibrosis (OR, 3.15; 95% CI, 1.25–7.93), fibrosis of any stage (OR, 1.66; 95% CI, 1.12–2.47), and NASH resolution (OR, 3.22; 95% CI, 2.17–4.7). Results were similar in patients without diabetes [209].

As a result, pioglitazone, together with vitamin E, is currently recommended for the treatment of MASLD with significant fibrosis.

### 4.3. Lipid-Lowering Drugs

#### 4.3.1. Statins

Statins are currently the most frequently used antilipidemic drugs for primary and secondary prevention of CVD, as the reduction in pro-atherogenic LDL cholesterol with statin therapy lowers the risk of major coronary events, coronary revascularization, ischemic stroke, and all-cause mortality [210]. Via inhibition of HMG-CoA reductase, statins inhibit cholesterol formation by targeting a key step in the biosynthesis of sterols and promoting lipoprotein clearance, particularly by inducing low-density lipoprotein receptor (LDL-R) expression [211].

Beyond cholesterol-lowering effects, statins exert pleiotropic anti-inflammatory, proapoptotic, anti-oxidative, and anti-fibrotic properties, which may prove beneficial for the improvement of MASLD/MASH [212]. This has already been proven in animal studies, where statins show a reduction in inflammation, fibrosis, hepatic stellate cell activation, and sinusoidal endothelial cell dysfunction [213,214,215].

To date, there are no RCTs evaluating the role of statins in MASH. Data from post hoc analyses of prospective studies demonstrate a beneficial trend of statin use on MASH. A post hoc analysis of the GREACE study highlighted statin safety, statin-induced improvement of liver tests, and reduction in cardiovascular risk in patients with established coronary heart disease and mild-to-moderately abnormal liver tests potentially attributable to MAFLD [216]. Another post hoc analysis of the IDEAL study proved the greater cardiovascular benefit of intensive statin therapy in contrast to moderate treatment strategy in patients with established cardiovascular disease and moderately elevated aminotransferase levels, emphasizing that moderate elevations in liver enzyme levels should not present a barrier to prescribing statins, even at higher doses, in high-risk patients [217]. In the post hoc analysis of the ATTEMPT study, which evaluated 1123 participants with metabolic syndrome without diabetes or CVD, liver enzymes and ultrasonography improved during the study with the use of statins [218]. Furthermore, improvement of hepatic steatosis has been observed in uncontrolled investigations using ultrasonography and tomography [219,220]. A histopathological follow-up study in patients with MASLD also revealed a significant reduction in liver steatosis with statin use [221]. A meta-analysis of 14 RCTs has highlighted the statins’ effect on significantly reducing liver biochemical indicators in patients with MASLD [222]. Moreover, statin use has been associated with a reduced risk of HCC development [223].

In December 2022, Ibrahim Ayada et al. published the results of a multidimensional study comprising a cross-sectional investigation in an ongoing general population cohort (the ROTTERDAM STUDY) and a MASLD patient cohort (the PERSONS cohort), a meta-analysis, and an experimental exploration. In the analysis of 4576 participants of the Rotterdam study, the use of statins in patients with dyslipidemia was inversely associated with MASLD compared to participants with untreated dyslipidemia (OR: 0.72, 95% CI: 0.59–0.86). In the PERSONS cohort, an analysis of 569 patients with biopsy-proven MASH showed that statin use was inversely associated with MASH (OR: 0.55, 95% CI: 0.32–0.95), but not with fibrosis (OR: 0.86, 95% CI: 0.44–1.68). The meta-analysis of 6 studies revealed a not significant inverse relation of statin use with steatosis (pooled OR: 0.69, 95% CI: 0.46–1.01), a significant inverse relation with MASH (pooled OR: 0.59, 95% CI: 0.44–0.79) and a significant inverse association with fibrosis (pooled OR: 0.48, 95% CI: 0.33–0.70). Lastly, experimental exploration in a fatty liver model of primary human liver organoids highlighted the hepatoprotective properties of statins by ameliorating the accumulation of intrahepatic lipid droplets and suppressing the expression of pro-inflammatory genes in macrophages [224].

Currently, STAT MASH (NCT04679376), a double-blind, placebo-controlled RCT, is being conducted to evaluate the safety and efficacy of atorvastatin in improving MASH features (improvement of MASH with no worsening of fibrosis or improvement of fibrosis with no worsening of MASH) in 70 adult patients with MASH.

In summary, although a beneficial trend of statins’ use in MASLD has been observed, stronger evidence from RCTs evaluating their possible positive effects in MASH patients is needed for definitive MASLD recommendations.

#### 4.3.2. Ezetimibe

Ezetimibe is an antilipidemic drug that lowers serum LDL levels via inhibition of Niemann Pick C1-like 1 protein (NPC1L1) expressed at the enterocyte/gut lumen (apical) as well as the hepatobiliary (canalicular) interface, thus reducing intestinal and biliary cholesterol absorption [225]. When added to statins, ezetimibe reduces LDL cholesterol levels by an additional 23 to 24%, on average. In IMPROVE-IT trial, combination of ezetimibe with statin reduced the risk of major adverse cardiovascular events (MACEs) compared with statin monotherapy [226]. Therefore, ezetimibe is currently used either in combination with statins in patients in whom statin monotherapy has failed to achieve the target LDL levels (according to individualized CV risk) or as monotherapy in patients with statin intolerance [227].

In experimental animal models, ezetimibe has been shown to ameliorate MASLD by improving liver steatosis and reducing IR [228,229]. Similar beneficial results have been observed in small-numbered human uncontrolled trials [230,231]. However, there is a lack of randomized, placebo-controlled clinical trials proving the beneficial effect of ezetimibe in MASLD. An open-label randomized clinical trial by Takeshita et al. demonstrated that treatment with 10 mg of ezetimibe for 6 months significantly reduced serum total cholesterol, fibrosis staging, and ballooning score, with no statistically significant effect on serum aminotransferase levels and hepatic steatosis. Using hepatic gene expression analysis, researchers suggested that amelioration of fibrosis might be regulated by the downregulation of genes involved in skeletal muscle development and cell adhesion molecules, thus suppressing stellate cell development into myofibroblasts. However, treatment with ezetimibe significantly deteriorated glycemic control, a critical risk factor for MASLD development and progression [232]. The MOZART trial, which assessed the efficacy of ezetimibe treatment in MASH patients by the magnetic resonance imaging-derived proton density-fat fraction (MRI-PDFF) and liver histology, failed to show a statistically significant reduction in liver fat content and serum aminotransferase levels [233]. Likewise, no improvement in liver steatosis was found in a meta-analysis by Lee and colleagues, despite the decrease in MASLD activity score [234]. In the ESSENTIAL study, open-label randomized trial of 70 MASLD patients assigned to receive either ezetimibe 10 mg plus rosuvastatin 5 mg daily or rosuvastatin 5 mg for up to 24 weeks, the combined ezetimibe plus rosuvastatin treatment significantly reduced liver fat measured by MRI-PDFF compared to single rosuvastatin therapy [235]. Overall, ezetimibe therapy is still controversial in patients with MASLD, and more studies are required to establish its efficacy.

#### 4.3.3. PCSK9 Inhibitors

Proprotein convertase subtilisin/kexin type 9 inhibitors (PCSK9i) are a novel therapeutic option for very high-risk cardiovascular patients who either cannot achieve LDL goal with statin-ezetimibe combination therapy or cannot tolerate statin therapy due to side effects [227].

PCSK9 binds to the LDL receptor, leading it to lysosomal degradation inside hepatocytes, thus reducing LDLR-mediated LDL reabsorption from circulation [236].

In the landmark FOURIER and ODYSSEY RCTs, the PCSK9i antibodies evolocumab and alirocumab were shown to reduce the risk of recurrent cardiovascular events in patients with established ASCVD and under statin therapy by effectively lowering LDL cholesterol levels [237,238].

Animal and clinical studies have demonstrated that high intrahepatic or circulating PCSK9 levels augment hepatic fatty acids and triglyceride storage, which may contribute to the pathogenesis of MASLD [239].

Although there is limited data from clinical trials using PCSK9i as a therapeutic option for MASLD, their results indicate a potential reduction in liver steatosis, inflammation, and fibrosis with a possible benefit on cardiovascular risk. A randomized study of 40 patients with heterozygous familial hyperlipidemia demonstrated complete amelioration of previously diagnosed MASLD after one year of treatment with a PCSK9i, with improvement in liver structure and a further reduction in CV risk [240]. Moreover, in a retrospective, chart review-based study, 8 of 11 patients with a radiologic diagnosis of MASLD treated with PCSK9i for any reason reached complete radiological resolution of MASLD with a statistically significant reduction in serum ALT levels [241].

#### 4.3.4. Other Hypolipidemic Agents

Fibrates activate the PPAR-α and are generally effective in lowering triglycerides and cholesterol levels [242]. Several RCTs have been conducted to evaluate the CV benefit of fibrates, with their efficacy confined to patients with severe hypertriglyceridemia (>500 mg/dL) as a statin add-on therapy [227]. Regarding MASLD, preclinical animal studies have demonstrated the positive effect of fibrates in ameliorating both hepatic steatosis and necroinflammation in MASLD or NASH experimental animals [243,244,245,246]. However, clinical-based research studies investigating the possible therapeutic benefit of fibrates in treating MASLD/MASH are limited. Fabbrini et al.’s study failed to show any alteration in intrahepatic triglyceride content in obese MASLDNAFLD patients treated either with fenofibrate or niacin [247]. Yaghoubi et al., on the contrary, demonstrated that fenofibrate significantly reduced serum transaminase levels, blood pressure, and BMI in 30 MASLD patients [248]. More clinical studies are therefore needed to establish the therapeutic effects of fibrates in MASLD treatment.

Ω-3 fatty acids [eicosapentaenoic acid (EPA) and docosahexaenoic acid (DHA)] are currently considered an effective add-on therapy to statins in high CV risk patients with elevated triglyceride levels despite statin treatment, in order to reduce the risk of ischemic CV events [249]. Clinical data also support the beneficial effects of ω-3 fatty acid supplementation in the decrease in hepatic steatosis in MASLD patients, rendering them a promising pharmacologic agent for the management of MASLD [250,251]; however, the size of the observed effect of ω−3 fatty acids is relatively small, and the optimal dose and duration of treatment are not yet established. Hence, additional RCTs are required before safe consumption can be made.

### 4.4. THR-β Agonists-Resmetirom

Thyroid hormone receptor alpha (THR-α) and beta (THR-β) are the main mediators of thyroid hormone effects into peripheral tissues. THR-β is the dominant receptor isoform in liver cells and regulates several metabolic pathways. The THR-β role is crucial in increasing fatty acid β-oxidation and lowering serum cholesterol and triglyceride levels by reducing the secretion of VLDL and upregulating expression of the LDLR in the hepatocytes [104]. However, THR-β hepatic signaling in patients with NASH is impaired, potentially worsening MASH and liver fibrosis [252].

Resmetirom, a liver-targeted THR-β selective agonist, showing promising results in the improvement of hepatic fibrosis and MASH resolution, has recently received FDA approval for non-cirrhotic MASH patients with moderate to advanced fibrosis, along with dietary and exercise interventions [104]. The resmetirom-MASH development program includes four ongoing phase 3 clinical trials: MAESTRO-NASH, MAESTRO-NAFLD-1, MAESTRO-NAFLD-OLE, and MAESTRO-NASH-OUTCOMES. Results were recently published from the MAESTRO-NASH trial. In the MAESTRO-NASH trial, a randomized, double-blind, placebo-controlled, phase 3 trial of 966 patients with biopsy-proven NASH and moderate or advanced liver fibrosis, MASH resolution with no worsening of fibrosis was observed in a statistically significantly higher number of patients in the resmetirom group when compared with the placebo one. Furthermore, the other primary endpoint of the trial, an improvement in fibrosis by at least one stage with no worsening of the MASLD activity score, was also achieved in significantly more patients in the resmetirom group [253]. Limitations of this study include the lack of clinical outcome data to correlate with histologic data and the unassessed long-term safety of resmetirom.

Another very interesting finding from the resmetirom trials comes from the MAESTRO-NAFLD trial, where resmetirom was shown to significantly lower atherogenic lipids/lipoproteins that contribute to increased CV risk, such as apoCIII, Lp(a), and VLDL cholesterol when compared to placebo [254]. Similar results were also observed in the MAESTRO-NASH trial [253].

### 4.5. Acetylsalicylic Acid (ASA)

Acetylsalicylic acid (ASA) irreversibly inhibits the cyclo-oxygenase 1 (COX1) enzyme, exerting anti-thrombotic effects via suppressing thromboxane A2 formation, a molecule that activates platelets and promotes vasoconstriction. ASA is used for secondary prevention in all patients with established atherosclerotic CVD, including chronic coronary disease, previous myocardial infarction, previous ischemic stroke, and peripheral arterial disease [255].

An in vitro and in vivo study has recently shown that aspirin administration in dyslipidemic conditions can simultaneously ameliorate nonalcoholic fatty liver and atherosclerosis through elevation of catabolic metabolism, inhibition of lipid synthesis, and suppression of inflammation, effects mediated from sequential regulation of the PPARδ-p-AMPK-PGC-1α oxidative phosphorylation pathway [256].

In a prospective cohort study that included 361 patients with biopsy-proven MASLD, administration of ASA on a daily basis was correlated with less severe histologic features of MASLD/MASH and reduced risk of progression to advanced fibrosis, especially after 4 years of aspirin use [257].

Additionally, daily aspirin therapy was significantly associated with a decreased risk for HCC in NAFLD patients in a population-based cohort study [258]. Recently, a randomized, double-blind, placebo-controlled phase 2 clinical trial with 80 non-cirrhotic MASLD patients tested as a primary endpoint whether aspirin use could reduce liver fat content. Researchers observed that a 6-month daily low-dose aspirin significantly reduced hepatic fat quantity measured by magnetic resonance imaging proton density fat fraction (MRI-PDFF) compared with placebo [259].

The latest clinical data suggest an underlying hepatoprotective effect of ASA on MASLD. Although this assumption needs further clarification, it seems to be an interesting approach, which can also prove beneficial in patients already requiring ASA for secondary cardiovascular protection.

**Table 2 biomolecules-15-00324-t002:** Summary of the clinical data about the effectiveness of cardiometabolic drugs in reducing hepatic steatosis, steatohepatitis, hepatic fibrosis, and cardiovascular risk. Abbreviations: ACE: Angiotensin Converting Enzyme; ARBs: Angiotensin Receptor Blockers; MASLD: metabolic dysfunction-associated steatotic liver disease; NASH: nonalcoholic steatohepatitis; MASH: metabolic dysfunction-associated steatohepatitis; HFrEF: heart failure with reduced ejection fraction; GLP-1: Glucagon-like Peptide-1; GIP: Glucose-Dependent Insulinotropic Peptide; PCSK9i: proprotein convertase subtilisin/kexin type 9 inhibitors. Strength of evidence (SoE): Ia: systematic reviews and/or meta-analyses of randomized clinical trials (RCTs) only from level Ib; Ib: phase 3, placebo-controlled, double-blinded RCTs with primary or secondary endpoint(s) for MASLD steatosis and/or steatohepatitis and/or fibrosis. IIa: systematic reviews and/or metanalyses including RCTs from level IIb; IIb: RCTs with primary or secondary endpoint(s) for MASLD steatosis and/or steatohepatitis and/or fibrosis not included in SoE Ib (e.g., phase 2 RCTs, single arm RCTs, open-label RCTs). IIIa: Systematic reviews of cohort studies, non-randomized clinical studies (NRCTs), case–control studies, or mixed studies (cohort studies, non-randomized CTs, case–control studies). IIIb: Nonrandomized RCTs, cohort studies, case–control studies. IV: case series, case-report studies, expert opinion.

Drug Category	Drug	Reduction in Hepatic Steatosis	Reduction in Steatohepatitis	Reduction in Hepatic Fibrosis	Assessment Methods	References	Cardiovascular (CV) Risk	Comments
Anti-hypertensives	RAAS inhibitors (ACE Inhibitors, ARBs)	Controversial (SoE: IIa)	Controversial (SoE: IIa)	Controversial (SoE: IIa)	Histology, liver ultrasound (US)	[117,118]	Reduce CV risk (used in patients with HTN, HFrEF, post-MI)	
	Mineralocorticoid Receptor Antagonists	Controversial (SoE: Ib—limited data)	N/A	Not Effective (SoE Ib—limited data)	MRI-based techniques, NAFLD liver fat score, APRI score	[130,132]	Reduce CV risk in specific conditions (e.g., HFrEF)	Potential benefit in MASLD liver fat score when combined with vitamin E
Glucose-lowering	GLP-1 Receptor Agonists	Effective (SoE: IIa)	Effective (SoE: IIa)	Not Effective (SoE: IIa)	Histology, MRI-based techniques, liver US	[157,158,159,160,161,162,163,164]	Reduce CV risk in high-risk patients with T2DM or obesity	Data in MASH patients without T2DM are scarce
	SGLT-2 Inhibitors	Effective (SoE: Ib)	N/A	Effective (LoE: IIa—quantitively limited data)	Histology, MRI-based techniques, Fibrosis-4 Index score	[179,180,181,183]	Proven cardiovascular benefits, including a reduction in heart failure hospitalizations and cardiovascular mortality in HFrEF patients	Limited data in non-diabetic MASLD
	Dual GLP-1/GIP Receptor Agonist	Limited evidence showing effectiveness (SoE: IIb)	N/A	N/A	MRI	[169]	Potential to reduce cardiovascular risk	Ongoing trials are evaluating effects on MASH, fibrosis, and cardiovascular outcomes
	Thiazolidinediones	Effective (E.S: Ia)	Effective (E.S: Ia)	Effective (E.S: Ia)	Histology and MRI-based techniques	[202,203,204,205,206,207,208,209]	Mixed cardiovascular effects May reduce CV risk in T2DM patients but can increase risk of heart failure	Recommended in combination with vitamin E for the treatment of MASH with significant fibrosis
	Metformin	Not Effective (SoE: IIa)	Not effective (SoE: IIa)	Not Effective (SoE IIa)	Histology and US	[192,193]	May reduce CV risk in T2DM	Recent meta-analyses do not support liver benefits Not recommended as a specific MASLD treatment
Lipid-lowering	Statins	Effective (SoE: IIb—limited data)	Effective (SoE: IIIa—limited data)	Effective (SoE IIIa—limited data)	Histology and US	[219,220,221,224]	Reduce CV risk	Fibrates
	Ezetimibe	Controversial (LoE: IIb—limited data)	Effective (LoE: IIIb—limited data)	Controversial (LoE: IIb—limited data)	Histology and MRI-based techniques	[230,231,232,233,234,235]	Reduce CV risk when combined with statins	May worsen glycemic control in some patients Combined ezetimibe-rosuvastatin treatment significantly reduces liver fat compared to rosuvastatin monotherapy
	PCSK9 inhibitors	Effective—(LoE: IIIb limited data)	N/A		US, Computed tomography (CT), MRI	[240,241]	Reduce CV events in high-risk patients when added to statin therapy	Positive trend towards hepatic steatosis and fibrosis amelioration
		N/A	N/A	N/A	N/A	N/A	Benefit in patients with severe hypertriglyceridemia when combined with statins	Preclinical studies suggest potential liver benefits
	Ω-3 fatty acids	Effective (LoE IIa)	Controversial (LoE IIIa)	Controversial (LoE IIIa)	US, MRI	[250,251]	May reduce cardiovascular risk	Data mainly from small, non-randomized trials
Others	ASA	Effective (LoE IIb—limited data)	Effective (LoE IIIa—limited data)	Effective (LoE: IIIa—quantitatively limited data)	MRI- based techniques, validated laboratory scores	[257,259]	Reduce CV risk	

## 5. Discussion

MASLD is independently associated with CVD occurrence; therefore, the CVD risk of patients with MASLD seems to be increased from the disease’s early stages. Special consideration should be made for patients with MASH or advanced fibrosis (F3-F4) as well as MASLD patients with concomitant T2DM, as these patients show an even higher CVD risk. The precise pathophysiological mechanisms linking MASLD with CVD are yet to be fully elucidated. Current evidence suggests an important role of insulin resistance along with other common risk factors, including dyslipidemia, IR, systemic inflammation, coagulopathies, ectopic adipose tissue deposition, perturbations in the gut microbiome, as well as a genetic component. Hepatokines, such as FGF-21 and Fetuin-A, have recently emerged as new players in the pathogenesis of both MASLD and CVD [260,261].

With a diagnosis of MASLD, a thorough cardiovascular risk assessment and evaluation for subclinical atherosclerosis should be considered to identify high-risk patients that are candidates for therapeutic interventions that alleviate the cardiometabolic risk factor burden.

Lifestyle modifications, including weight loss, augmented physical activity, and suitable nutrition, e.g., Mediterranean/low-carbohydrate diet, which both lead to weight loss, remain the cornerstone of MASLD management [262]. New agents targeting explored pathways implicated in MASLD progression are currently tested in preclinical and randomized clinical trials [263]. Interestingly, FGF-21 analogs have been tested in phase 1 and phase 2 studies and provided favorable effects on MASLD progression [264].

However, pharmacologic interventions with drugs targeting known modifiable cardiometabolic risk factors, such as hypertension, dyslipidemia, and T2DM, remain a reasonable and common strategy to prevent CVD development and progression of MASLD. Anti-hypertensive agents haven’t yielded beneficial effects on MASLD progression. Metformin remains the first-line treatment for T2DM and the most widely used regimen among patients with T2DM. Although insulin resistance plays an important role in the pathogenesis of MASLD, metformin failed to prove beneficial effects on the MASLD progression, in contrast with the other category of insulin sensitizers, thiazolidinediones, which in RCTs have shown improvement in steatohepatitis while delaying the progression of liver fibrosis.

RCTs designed to primarily prove the efficacy of GLP-1Ra to improve glycemic control and/or to reduce body weight in T2DM and/or obese patients suggested favorable effects in steatosis and steatohepatitis progression, opening promising therapeutic avenues [265]. However, not all studies have demonstrated improvement of liver fibrosis [161,162].

SGLT-2i is another class of glucose-lowering agents that gathered interest as a new weapon for the management of MASLD. SGLT-2i inhibitors seem to exert pleiotropic actions, regulating pathways crucial for the MASLD development and progression [266]. However, their effects have been studied in open-label studies yielding conflicting results [267,268].

Regarding the lipid-lowering drugs, statins, either alone or combined with ezetimibe, have shown improvement in liver steatosis. Of note, an RCT study evaluating the effects of statins on MASH and fibrosis is now ongoing. ASA emerges as another promising drug for MASLD, since, albeit limited, the existing data demonstrated improvement of all stages of MASLD progression. Recently, a novel drug for MASH, resmetirom, has shown positive effects regarding CVD risk, opening new opportunities for the therapeutic approach of MASLD and CVD (Figure 2, Table 2).

CVD remains the cardinal cause of mortality in people with MASLD, while those with both MASLD and T2DM are in significantly higher risk of death. Increasing clinical interest has been focused on reversing the advanced stages of fibrosis (≥3), since these are the ones that are associated with increased overall mortality [269]. Large, high-quality, randomized-controlled trials aiming to assess by biopsy the efficacy of the various cardiometabolic drugs to delay/reverse MASLD progression, as a primary outcome, are lacking. New lipid-lowering and anti-hyperglycemic agents such as PCSK-9i, tirzepatide, and triple agonists (GLP-1/GIP/glucagon receptor triagonist) as well as the combination of cardiometabolic agents of different categories could be evaluated for their effects on MASLD progression in an attempt to lessen the patient and economic burden of the disease.

## Figures and Tables

**Figure 1 biomolecules-15-00324-f001:**
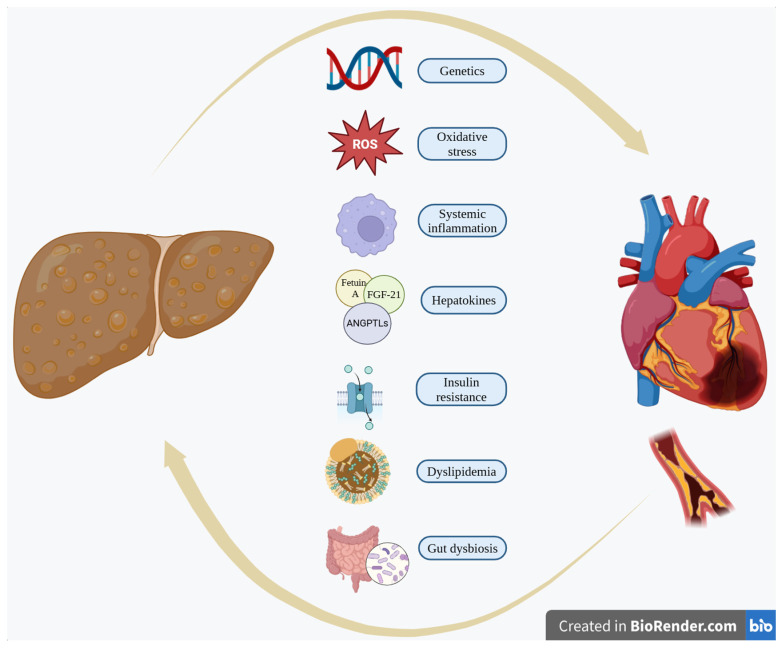
Proposed pathophysiologic mechanisms connecting MASLD and CVD.

**Figure 2 biomolecules-15-00324-f002:**
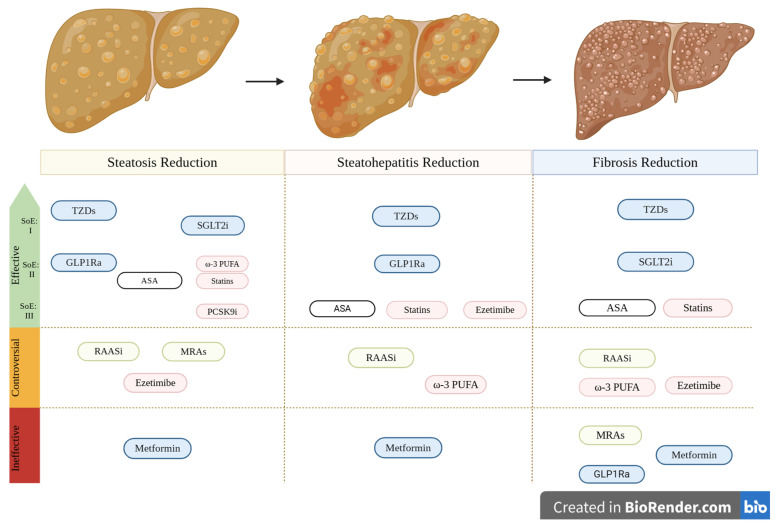
Potential cardiometabolic treatment options as therapeutic targets for MASLD and their effectiveness in liver steatosis, steatohepatitis, and liver fibrosis reduction. Abbreviations: MASLD: metabolic dysfunction-associated steatotic liver disease; RAASi: Renin–Angiotensin–Aldosterone system inhibitors; MRAs: Mineralocorticoid Receptor Antagonists; GLP1Ra: Glucagon-like Peptide-1 Receptor Agonists; SGLT2i: sodium glucose transporter-2 inhibitors; TZDs: thiazolidinediones; PCSK9i: proprotein convertase subtilisin/kexin type 9 inhibitors; ω-3 PUFA: omega-3 polyunsaturated fatty acid; ASA: acetylsalicilic acid; SoE: strength of evidence. Notes: glucose-lowering drugs are illustrated with blue color, anti-hypertensive drugs with green color, and lipid-lowering drugs with pink color. Strength of evidence (SoE): Ia: systematic reviews and/or meta-analyses of randomized clinical trials (RCTs) only from level Ib; Ib: phase 3, placebo-controlled, double-blinded RCTs with primary or secondary endpoint(s) for MASLD steatosis and/or steatohepatitis and/or fibrosis. IIa: systematic reviews and/or meta-analyses including RCTs from level IIb; IIb: RCTs with primary or secondary endpoint(s) for MASLD steatosis and/or steatohepatitis and/or fibrosis not included in SoE Ib (e.g., phase 2 RCTs, single-arm RCTs, open-label RCTs).

## Data Availability

Not applicable.

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
