# Peer review of "Linking Cardiovascular Disease and Metabolic Dysfunction-Associated Steatotic Liver Disease (MASLD): The Role of Cardiometabolic Drugs in MASLD Treatment"

_biomolecules, 2025, doi:10.3390/biom15030324_

Round 1
Reviewer 1 Report
Comments and Suggestions for Authors
This paper provides a nice overview of many of the current drugs being used to treat patients with MASLD. Please address the following points to improve quality and significance.
1. Simplify the title using only one disease designations. This reviewer recommends using MASLD. Related to this, the authors should choose one disease designation and use that in the paper. Switching back and forth between NAFLD, NASH, MASLD, etc. is not optimal. Again, this reviewer suggests that the authors stick with MASLD as this is a “Catch-All” description.
2. Line 30, MASH is not defined in the abstract
3. Section 2: This section would be improved by providing more specific descriptions of the “types” of CVD associated with MASLD. There are too many distinct types of CVD to simply use the blanked term CVD.
4. Section: Adding a section of hepatokines is important to further make connections between MASLD and CVD.
5. Line 329 – Define ARBs
6. The paper ends abruptly. The authors might consider providing a summary figure where the “rate” the effectiveness of the different drugs discussed in treating MASLD and CVD. Alternatively, this information could be added into the Figure on page 18. This would help the readers.
Reviewer 2 Report
Comments and Suggestions for Authors
This manuscript reviews the effect of cardiovascular drugs on some hepatic diseases.
I consider that this review can be of interest as it gives some insight that can be useful when treating patients with these problems.
Some things to consider:
- The authors indicate in the introductions that MAFLD is related to metabolic syndrome. This explains the need to use cardiovascular drugs in these patients. As this is an important point, I suggest that the authors explain a little longer this relation and the importance of managing cardiovascular drugs.
- Page 9 The title of section 4 is missing. It seems that a paragraph is missing.
- I consider that Figure 2 is not necessary, it does not give any important information. Moreover, the term anti-lipidemics should be changed by lipid lowering agents.
- A discussion sections should be included
- The title of section 6 is misleading.
Author Response
Please see attatcment

Reviewer 3 Report
Comments and Suggestions for Authors
· Abstract:
- The use of unnecessary abbreviations in the abstract detracts from its readability and should be eliminated.
- The abstract lacks informativeness as it does not present any results or key findings from the review. Including specific outcomes would enhance its utility.
- Does "salicylic acid" refer to acetylsalicylic acid?
· Scope and Focus:
- The review’s stated topic is the effects of cardiometabolic drugs on liver disease. However, sections 2 and 3 (spanning pages 3 to 8) primarily discuss the epidemiological and pathophysiological links between MAFLD and cardiovascular disease. While this information is relevant, it would be more appropriate as part of the introduction. These sections should either be removed or significantly condensed to align with the main focus of the review.
· Figures and Visual Quality:
- Figures 1 and 2 are of substandard quality and need improvement to enhance their clarity and visual appeal.
- Figure 2, in particular, lacks informativeness. For most interventions depicted, robust evidence regarding their effects on liver disease is unavailable, reducing the figure’s utility.
· Focus on Clinical Data:
- The review should place greater emphasis on clinical data rather than uncertain pathophysiological effects of the interventions. A stronger focus on clinically validated outcomes would significantly improve its relevance and impact.
· Tabular Representation:
- Constructing a table that presents existing evidence focusing exclusively on hard clinical outcomes is strongly recommended.
- A secondary table could summarize the effects of the evaluated drugs on biomarkers, providing a clearer synthesis of the available data.
· Content Gaps:
- Chronic kidney disease, a key component of cardiometabolic health, has been completely overlooked in the review. This omission should be addressed to provide a more comprehensive analysis.
· Guidance for Future Research:
- The review should include a dedicated section offering guidance for future research. Highlighting areas where evidence is lacking or where further studies are needed would add significant value.
Comments on the Quality of English Language
Minor revisions needed
Round 2
Reviewer 2 Report
Comments and Suggestions for Authors
The authors have followed the indications of the review. They have made important improvements and different changes in the manuscript
Author Response
Dear Reviewer,
Thank you very much for appreciating all the changes/ improvement that we made to our manuscript according to your precious suggestions.
Sincerely yours
Prof Eva Kasi